# Pattern and correlates of physical activity and sedentary behaviours of pregnant women in Ibadan, Nigeria: Findings from Ibadan pregnancy cohort study

Ikeola A. Adeoye [1,2] *

**1** Department of Epidemiology and Medical Statistics, College of Medicine, University of Ibadan, Ibadan, Nigeria, **2** Consortium for Advanced Research in Africa (CARTA), Nairobi, Kenya

* adeoyeikeola@yahoo.com

**Data Availability Statement:** The datasets generated and/or analysed during the current study which is ongoing are not publicly available because they contain potentially identifying and confidential

## Abstract

Globally, physical inactivity is the fourth leading risk factor for premature death. Pregnancy is associated with reduced physical activity because of physiological and anatomical changes and socio-cultural barriers. Even though physical activity provides many benefits, such as improved insulin sensitivity and reduced cardiometabolic risk, it is not emphasized among pregnant women in Nigeria. This study described the pattern of physical activity and sedentary behaviours of pregnant women from the Ibadan Pregnancy Cohort Study in Ibadan, Nigeria. The Ibadan Pregnancy Cohort Study (IbPCS) is a prospective cohort study investigating the associations between maternal obesity, lifestyle factors on glycaemia control, gestational weight gain, pregnancy and postpartum outcomes among pregnant women in Ibadan. The Pregnancy Physical Activity Questionnaire (PPAQ) was used to assess physical activity and sedentary behaviour. Sedentary time was estimated from the time spent watching television, sitting at work and the computer. Bivariate and multivariate logistic regression analyses were done to investigate associations at a 5% level of statistical significance. None of the pregnant women met the WHO recommendation of 150mins of moderate-intensity activity per week. The average time spent engaged in moderate-intensity activity was 26.3 ± 22.9 mins. The mean daily sedentary time was 6.5 ± 4.2 hours. High parity para ≥ 4: [AOR 0.57 95% CI: (0.36–0.89) p = 0.014] and being employed [AOR 0.23 95% CI: (0.15–0.33) p <0.001] reduced the odds of having inadequate physical activity. Correlates of sedentary behavior after adjusting for confounders were high parity: para 1–3 AOR 0.73, 95% CI: (0.58–0.91) p = 0.004], tertiary education: AOR 2.39 95% CI: (1.16–4.91) p = 0.018] and earning a higher income: AOR 1.40: 95% CI: (1.11–1.78) p = 0.005]. Pregnant women's physical activity and sedentary behaviours are emerging public health issues, especially in Nigeria. The level of physical activity was inadequate among pregnant women, while the sedentary time was high. There is a need to implement programmes that promote physical activity and discourage sedentary behaviour among pregnant women in Nigeria.

information but are available from the UI/UCH Ethics Committee (uiuchec@gmail.com) on reasonable request if it meets the criteria for accessing confidential data.

**Funding:** This research was supported by the Consortium for Advanced Research Training in Africa (CARTA). CARTA is jointly led by the African Population and Health Research Center and the University of the Witwatersrand and funded by the Carnegie Corporation of New York (Grant No. G-19-57145), Sida (Grant No:54100113), Uppsala Monitoring Center, Norwegian Agency for Development Cooperation (Norad), and by the Wellcome Trust [reference no. 107768/Z/15/Z] and the UK Foreign, Commonwealth & Development Office, with support from the Developing Excellence in Leadership, Training and Science in Africa (DELTAS Africa) programme. The statements made and views expressed are solely the responsibility of the Fellow. Ikeola Adeoye is a CARTA PhD fellow. The funders had no role in study design, data collection and analysis, decision to publish, or preparation of the manuscript. For the purpose of open access, the author has applied a CC BY public copyright license to any Author Accepted Manuscript version arising from this submission.

**Competing interests:** The author has declared that no competing interests exist.

## Introduction

Physical inactivity (PA) is globally the number four leading risk factor for premature death [1,2]. The current obesity epidemic has drawn attention to the importance of physical activity among the general population. Regular physical activity is beneficial throughout the life course because it improves cardiorespiratory fitness, reduces the risk of obesity, diabetes, cardiovascular diseases, and certain cancers, and prolongs life [3]. Additionally, during pregnancy, physical activity improves insulin sensitivity and glucose uptake; hence can prevent cardio-metabolic outcomes such as gestational diabetes mellitus and excessive gestational weight gain [4–6]. The other benefits of physical activity during pregnancy include improved cardiovascular fitness, sleep quality, and psychological well-being [7]. Generally, pregnancy is associated with reduced physical activity because of physiological and anatomical changes. For example, increased lumbar lordosis, weight gain and fatigue which affects the ease, willingness and safety involved in physical activity, particularly as the pregnancy progresses [8]. The other reasons for reduced physical activity include socio-cultural beliefs, misinformation, and the fear of complications [9]. Also, pregnant women are often encouraged to rest rather than exercise because of the presumed maternal and foetal complications, such as miscarriage, preterm delivery and intrauterine growth restriction that results from reduced placenta circulation. Exercise is thought to shunt blood away from the placenta and other vital organs to the skeletal muscles during exercise [10].

Physical activity is defined "as any bodily movement produced by skeletal muscles that result in energy expenditure" [11]. The degree of energy expenditure is influenced by the intensity, duration, and frequency of muscular contractions and muscle mass. Physical activity is classified according to types, which have been described in four distinct domains: occupational, household task, leisure-time, e.g. sports, and transport [12]. It could also be categorized by intensity (light, moderate or heavy intensity) or time of the week (weekday and weekend). In contrast, exercise is a planned, structured, repetitive physical activity to improve physical fitness. The types of exercise are aerobic exercises (e.g. walking, which increases strength and cardiovascular fitness), resistance exercises (which increase muscle mass and strength), and stretching exercises (which improve flexibility by increasing muscle fibre size) [11].

The World Health Organization and some allied professional organizations, such as the American Congress of Obstetricians and Gynaecologists (ACOG), and Royal College of Obstetricians and Gynaecologists (RCOG), among others, have formulated physical activity guidelines during pregnancy [1,2,13]. These guidelines are evidence-based recommendations for practising physical activity during pregnancy to ensure the health and safety of the mother, foetus and neonate. The recommendation of WHO is that pregnant women should engage in at least 150 minutes of moderate-intensity physical activity during the week. Similarly, the ACOG advocated that "in the absence of either medical or obstetric contraindications, 30 min or more of moderate exercise a day on most, if not all, days is recommended for pregnant women" [13]. In 2015, ACOG revised the earlier recommendation based on the current scientific data [1], which still affirmed the previous guidance of 30 minutes of moderate-intensity exercise on most days of the week. Unfortunately, physical activity among pregnant women has received much less attention in sub-Saharan Africa, including Nigeria [14]. Physical activity is not yet an essential component of maternal health services in Nigeria regarding available policy, guidelines, counselling and recommendation.

Sedentary behaviour (SB) has been defined as "any waking behaviour characterized by an energy expenditure $\leq 1.5$ metabolic equivalents (METs) while sitting or reclining position" [15,16]. At the same time, sedentary time (ST) is the time spent for any duration in sedentary behaviour [15]. Too much time spent in sedentary behaviour compromises metabolic health

and is a significant risk factor for obesity, type 2 diabetes, cardiovascular disease and all-cause mortality [17–19]. The Canadian Society for Exercise Physiology recommends that sedentary time should be less than 8 hours (with less than 3 hours of recreational screen time [20]. Sedentary behaviour differs from physical inactivity in that individuals who meet physical activity recommendations are at higher risk of adverse metabolic complications if much time is spent on sedentary behaviour [19,21]. Sedentary behaviour during pregnancy is associated with adverse outcomes, for example, gestational weight gain [22], gestational diabetes mellitus [23], hypertensive disorders [24] and macrosomia [25]. Pregnant women's physical activity and sedentary behaviours are emerging public health issues, especially in Sub-Saharan Africa. This results from epidemiologic and nutritional transitions characterized by changes in dietary patterns, reduced physical activity and increased sedentary behaviours [26–29] particularly in the cities. This is a departure from the traditional African culture in which women were physically active during pregnancy, particularly the rural dwelling women of Africa belonging to the lower socioeconomic status [30]. While there is a paucity of studies assessing physical activity among pregnant women in Nigeria, information on sedentary behaviour among pregnant is lacking in Nigeria. Therefore, this study evaluated the pattern and the correlates of physical activity and sedentary behaviour among pregnant women in Ibadan, Nigeria, using the Ibadan Pregnancy Cohort Study.

## Materials and methods

### Study design and participants

The Ibadan Pregnancy Cohort Study (IbPCS) is a prospective cohort study investigating the associations between maternal obesity, lifestyle factors on glycaemic control, gestational weight gain, pregnancy and postpartum outcomes among pregnant women in Ibadan. The lifestyle factors examined included dietary patterns, sugar-sweetened beverages, physical activity, sedentary behaviour, tobacco use, alcohol consumption, and sleep patterns. The study was conducted at four medical facilities within the Ibadan metropolis: University College Hospital, Adeoyo Maternity Teaching Hospital, Jericho Specialist Hospital, Saint Mary Catholic Hospital, Oluyoro, Ibadan, Oyo State, Nigeria. These hospitals offer comprehensive obstetric services to pregnant women and are the major referral centres in the city. IbPCS recruited 1745 pregnant women in early pregnancy ($\leq$ 20 weeks' gestation) during their antenatal booking visit and followed them until delivery. Data were collected using pretested, interviewer-administered questionnaires and desktop review of medical records at three points during the study–booking, third trimester, and delivery. The level of physical activity and sedentary behaviours were assessed at baseline and other parameters described in the protocol [31]. The details of the methodology have been published elsewhere. Physical activity (PA) and sedentary behaviour were assessed using the Pregnancy Physical Activity Questionnaire (PPAQ). Sedentary time was estimated from the number of hours spent sitting in a day: i.e. the number of hours spent on average watching television per day and on weekends, sitting at the computer, and the number of hours spent sitting at work daily.

**Assessment of physical activity.** The Pregnancy Physical Activity Questionnaire (PPAQ) assessed physical activity patterns and levels. The PPAQ was developed and validated by Chasan-Taber in 2004 to determine the physical activity levels in pregnant women [32]. It is a 32-item questionnaire that measures the levels of physical activity across five domains–Household and caregiving activities (13 items), occupational (5 items), sports and exercise (12 items), and transport (3 items). The analysis of physical activity was done according to the PPAQ instruction guide [32]. In summary, the activity assessed was classified by intensity (sedentary, light, moderate and vigorous) and by the pattern of activity (household/care,

occupation, transport, sports). Responses to each question were reckoned in time: none, less than ½ hours per day, ½ to almost 1 hour per day, 1 to nearly 2 hours per day, 2 to almost 3 hours per day, and three or more hours per day. Energy expenditure (MET-hours per week) was estimated by multiplying the mid-point of the time option by the metabolic equivalent allotted to the activity. The metabolic equivalent (MET) is the ratio of a person's working metabolic rate relative to the resting metabolic rate. The total energy expenditure was obtained by summing all activities' energy expenditure. Total physical activity was categorized as inadequate if the total score was within the 50th percentile and high if it was > the 50th percentile. Factors associated with inadequate physical activity were assessed.

**Assessment of sedentary behaviour.** Sedentary behaviour was also assessed using the PPAQ according to the PPAQ instruction guide. Sedentary behaviour was calculated as the sum of the product of intensity and duration (MET-h/day) of sedentary activities (questions #11, 12, 13, 22, 32). Duration of moderate-intensity exercise (minutes per week) is the summation of time spent on moderate-intensity activity per week. Sedentary time (ST) was estimated by the time reportedly spent (in hours) watching television, sitting at the computer, and sitting at work. An eight-hour threshold is the recommended limit for ST within 24 hours [33].

## Data analysis

Statistical analysis was performed using STATA 13. Univariate analysis was used to describe the intensity and types of physical activity during pregnancy. Boxplots displayed the duration of moderate-intensity activity per week, and the pie chart shows the proportion that exceeds the recommended threshold for ST. A Chi-square test was performed to assess the relationship between the women's characteristics and total physical activity in tertiles. The outcome variables: total physical activity and sedentary behaviour, were categorized into binary variables (Inadequate = 1 and high = 0) using the cut-off of the 50th percentile. The explanatory variables included demographic variables, BMI and having motorized transport. Factors associated with inadequate physical activity were investigated using bivariate logistic analysis. Variables significant at a 5% level of statistical significance at bivariate logistic analysis (parity, employment status, religion, ownership of a motorized transport) were subjected to multiple logistic regression analysis. Unadjusted odds ratios (UOR), adjusted odds ratios (AOR), 95% confidence intervals and p-values ($p < 0.05$) were reported.

## Ethical consideration

The ethical approval for this study was obtained from the University of Ibadan/University College Hospital (UI/UCH) Institutional Review Board (UI/EC/15/0060) and Oyo State Ministry of Health Ethical Committee (AD/13/479/710). Both verbal and written informed consent was obtained from respondents before recruitment into the study. The study protocol and conduct adhered to the principles in the Declaration of Helsinki.

## Results

### Characteristics of pregnant women according to level of physical activity (Table 1)

A total of 1745 pregnant women were recruited for this study. The characteristics of pregnant women according to tertiles of total physical activity in the Ibadan Pregnancy Cohort Study are shown in Table 1. Maternal age (p = 0.007), parity ($p < 0.001$), marital status (p = 0.005), maternal education (p = 0.019), occupation ($p < 0.001$), ownership of motorized transport (p = 0.009) were associated with the total physical activity. Specifically, older women 40 years

**Table 1. Characteristics of pregnant women according to tertiles of total physical activity in the Ibadan Pregnancy Cohort Study.**

| Characteristics | Total | Tertiles of Total Physical Activity | | | Chi-square | p-value |
|---|---|---|---|---|---|---|
| | | Low | Middle | High | | |
| **Age group** | | | | | | |
| < 20 | 33 (1.9) | 11 (33.3) | 15(45.5) | 7(21.2) | 17.66 | **0.007** |
| 20–29 | 832 (47.7) | 299(36.0) | 271(33.0) | 262(32.0) | | |
| 30–39 | 812 (46.5) | 240(30.0) | 276(34.0) | 296(36.5) | | |
| ≥ 40 years | 68 (3.9) | 32(47.1) | 20(29.4) | 16(24.0) | | |
| **Parity** | | | | | | |
| Nulliparous | 760 (43.7) | 304 (40.0) | 256 (33.7) | 200 (26.3) | 40.83 | **<0.001** |
| 2–4 | 882 (50.8) | 246 (27.9) | 300 (34.0) | 336 (38.1) | | |
| ≥ 5 | 95 (5.5) | 29 (30.5) | 24 (25.6) | 42 (44.2) | | |
| **Marital Status** | | | | | | |
| Single | 102 (5.8) | 48 (47.1) | 32 (31.4) | 20 (21.6) | 10.70 | **0.005** |
| Married | 1643 (94.2) | 534 (32.5) | 550 (33.5) | 559 (34.0) | | |
| **Maternal Education** | | | | | | |
| Primary or less | 49 (2.8) | 13 (26.5) | 19 (38.8) | 17 (34.7) | 11.82 | **0.019** |
| Secondary | 504 (28.9) | 180 (35.7) | 185 (36.7) | 139 (27.6) | | |
| Tertiary | 1188 (68.2) | 387 (32.6) | 377 (31.7) | 424 (35.7) | | |
| **Occupation** | | | | | | |
| Employed | 1556 (89.2) | 453(29.1) | 542(34.8) | 561 (36.1) | 119.63 | **<0.001** |
| Unemployed | 186 (10.8) | 129 (68.3) | 40 (21.2) | 20 (10.6) | | |
| **Religion** | | | | | | |
| Christianity | 1010 (58.2) | 321 (31.8) | 332 (32.9) | 357 (35.4) | 5.20 | 0.074 |
| Islam | 726 (41.8) | 260 (35.8) | 245 (33.8) | 221 (30.4) | | |
| **Ethnicity** | | | | | | |
| Yorubas | 1564 (89.8) | 521 (33.3) | 534 (34.1) | 509 (32.5) | 5.94 | 0.051 |
| Non-Yorubas | 178 (10.2) | 61 (34.3) | 46 (25.8) | 71 (39.9) | | |
| **Income per month (Naira)** | | | | | | |
| <20,000 | 583 (38.0) | 189 (32.4) | 200 (34.3) | 194 (33.3) | 4.45 | 0.348 |
| 20,000–99,999 | 843 (55.0) | 234 (27.7) | 293 (34.8) | 316 (37.5) | | |
| ≥ 100,000 | 108 (7.0) | 30 (27.8) | 39 (36.1) | 39 (36.1) | | |
| **Body Mass Index (kg/m$^2$)** | | | | | | |
| Underweight | 50 (3.0) | 21 (42.0) | 20 (40.0) | 9 (18.0) | 11.65 | 0.070 |
| Normal weight | 845 (49.8) | 294 (34.8) | 267 (31.6) | 284 (33.6) | | |
| Overweight | 473 (27.9) | 151 (31.9) | 154 (32.6) | 168 (35.5) | | |
| Obese | 328 (19.3) | 98 (29.9) | 126 (38.4) | 104 (31.7) | | |
| **Motorized Transport** | | | | | | |
| **Yes** | 885 (50.7) | 264 (30.7) | 218 (32.7) | 315 (36.6) | 9.47 | **0.009** |
| **No** | 860 (49.3) | 318 (35.9) | 301 (34.0) | 266 (30.1) | | |

and above (47.1%) compared with younger women less than 20 years (33.3%), as well as unemployed women (68.3%) compared with women with gainful employment (29.1%) reported lower levels of physical activity. However, physical activity levels increased significantly with parity; (nulliparous: 26.3%), (1–3: 38.1%) (≥ 4: 44.2%) in a dose-response fashion. Also, married women (34.0%) had higher physical activity levels than single women (21.6%). Women with motorized transport (36.6%) also reported significantly higher physical activity levels than those without one (30.1%).

## Factors associated with inadequate physical activity among respondents (Table 2)

Table 2 shows the factors associated with inadequate physical activity among pregnant women in the Ibadan Pregnancy Cohort Study. By crude and adjusted logistic models, positive associations were found between parity, religion and occupational status and inadequate total physical activity. The odds of being less physically active decreased with increasing parity: para 1–3: AOR 0.67 95% CI: (0.55–0.82) p <0.001 and para ≥ 4: AOR 0.57 95% CI: (0.37–0.89) p = 0.014 compared with nulliparous women after adjusting for other factors. Additionally, employed women with AOR 0.23 95% CI: (0.16–0.34) p <0.001 had a lower likelihood of being less physically active than women that are unemployed.

**Table 2. Factors associated with inadequate physical activity level among the respondents.**

| Characteristics | Unadjusted OR (95% CI) | p-value | Adjusted OR (95% CI) | p-value |
|---|---|---|---|---|
| **Age group** | | | | |
| < 20 | 1.00 | | | |
| 20–29 | 0.85 (0.42–1.72) | 0.654 | | |
| 30–39 | 0.61 (0.30–1.23) | 0.170 | | |
| ≥ 40 years | 1.05 (0.45–2.44) | 0.905 | | |
| **Parity** | | | | |
| Nulliparous | 1.00 | | 1.00 | |
| 1–3 | 0.59 (0.48–0.71) | <0.001 | 0.67 (0.55–0.82) | **<0.001** |
| ≥ 4 | 0.51 (0.33–0.79) | 0.002 | 0.57 (0.36–0.89) | **0.014** |
| **Marital Status** | | | | |
| Single | 1.00 | | | |
| Married | 0.69 (0.46–1.03) | 0.068 | | |
| **Education** | | | | |
| Primary or less | 1.00 | | | |
| Secondary | 1.11 (0.62–2.01) | 0.713 | | |
| Tertiary | 0.90 (0.51–1.59) | 0.719 | | |
| **Occupation** | | | | |
| Employed | 0.20 (0.14–0.30) | <0.001 | 0.23 (0.16–0.34) | **<0.001** |
| Unemployed | 1.00 | | 1.00 | |
| **Religion** | | | | |
| Christianity | 1.00 | | 1.00 | |
| Islam | 1.27 (1.06–1.55) | 0.011 | 1.30 (1.07–1.59) | **0.009** |
| **Ethnicity** | | | | |
| Yorubas | 1.19 (0.88–1.63) | 0.262 | | |
| Non-Yorubas | 1.00 | | | |
| **Income per month (Naira)** | | | | |
| <20,000 | 1.00 | | | |
| 20,000–99,999 | 0.83 (0.67–1.02) | 0.087 | | |
| ≥ 100,000 | 0.68 (0.45–1.04) | 0.081 | | |
| **Body Mass Index (kg/m$^2$)** | | | | |
| Underweight | 1.00 | | | |
| Normal weight | 0.96(0.55–1.72) | 0.917 | | |
| Overweight | 0.80 (0.45–1.44) | 0.460 | | |
| Obese | 1.02 (0.56–1.85) | 0.954 | | |
| **Motorized transport** | 0.79 (0.66–0.96) | 0.016 | 0.88 (0.72–1.07) | 0.188 |

**Table 3. Factors associated with sedentary behaviour among pregnant women in Ibadan.**

| Characteristics | Unadjusted odds ratio (95% CI) | p-value | Adjusted odds ratio (95% CI) | p-value |
|---|---|---|---|---|
| **Age in years** | | | | |
| < 20 | 1 | | | |
| 20–29 | 1.11 (0.51–2.43) | 0.790 | | |
| 30–39 | 0.90 (0.42–1.99) | 0.809 | | |
| ≥ 40 years | 0.76 (0.30–1.92) | 0.569 | | |
| **Parity** | | | | |
| Nulliparous | 1 | | 1 | |
| 2–4 | 0.70 (0.56–0.86) | 0.001 | 0.73 (0.58–0.91) | **0.006** |
| ≥ 5 | 0.57 (0.35–0.88) | 0.011 | 0.81 (0.49–1.33) | 0.398 |
| **Marital Status** | | | | |
| Single | 1 | | | |
| Married | 0.73 (0.45–1.16) | 0.182 | | |
| **Education** | | | | |
| Primary or less | 1 | | 1 | |
| Secondary | 1.31 (0.69–2.52) | 0.421 | 1.26 (0.61–2.58 | 0.536 |
| Tertiary | 2.84 (1.49–5.40) | 0.001 | 2.39 (1.16–4.90) | **0.018** |
| **Occupation** | | | | |
| Employed | 0.63 (0.36–1.09) | 0.099 | | |
| Unemployed | 1 | | | |
| **Religion** | | | | |
| Christianity | 1 | | 1 | |
| Islam | 0.70 (0.57–0.86) | 0.001 | 0.87 (0.69–1.0) | 0.221 |
| **Income per month (Naira)** | | | | |
| <20,000 | 1 | | 1 | |
| 20,000–99,999 | 1.72 (1.38–2.16) | <0.001 | 1.41 (1.11–1.79) | **0.005** |
| ≥ 100,000 | 1.78 (1.17–2.75) | 0.008 | 1.31 (0.69–2.52) | 0.177 |
| **Body mass index** | | | | |
| Underweight | 1 | | | |
| Normal weight | 0.79 (0.42–1.46) | 0.444 | | |
| Overweight | 0.84 (0.45–1.59) | 0.597 | | |
| Obese | 0.77 (0.40–1.47) | 0.427 | | |

## Factors associated with sedentary behaviour among pregnant women (Table 3)

Table 3 shows the factors associated with sedentary behaviour among pregnant women in Ibadan with the unadjusted and adjusted odds ratios and 95% confidence intervals. On univariate analysis, high parity, tertiary education, employment status, religion and income were significantly associated with sedentary behaviour. The adjusted analysis showed that women with higher parity had lower odds for sedentary behaviour: para 1–3 AOR 0.73, 95% CI: (0.58–0.91) p = 0.004 compared with nulliparous women after adjusting for confounders. Women with tertiary education also had a higher likelihood of sedentary behaviour AOR 2.39 95% CI: (1.16–4.91) p = 0.018 compared with women with primary education. Earning a higher income also increased the odds of sedentary behaviour: "20,000–99,999" naira: 1.40: 95% CI: (1.11–1.78) p = 0.005 compared with lower-income earning women (<20,000.00).

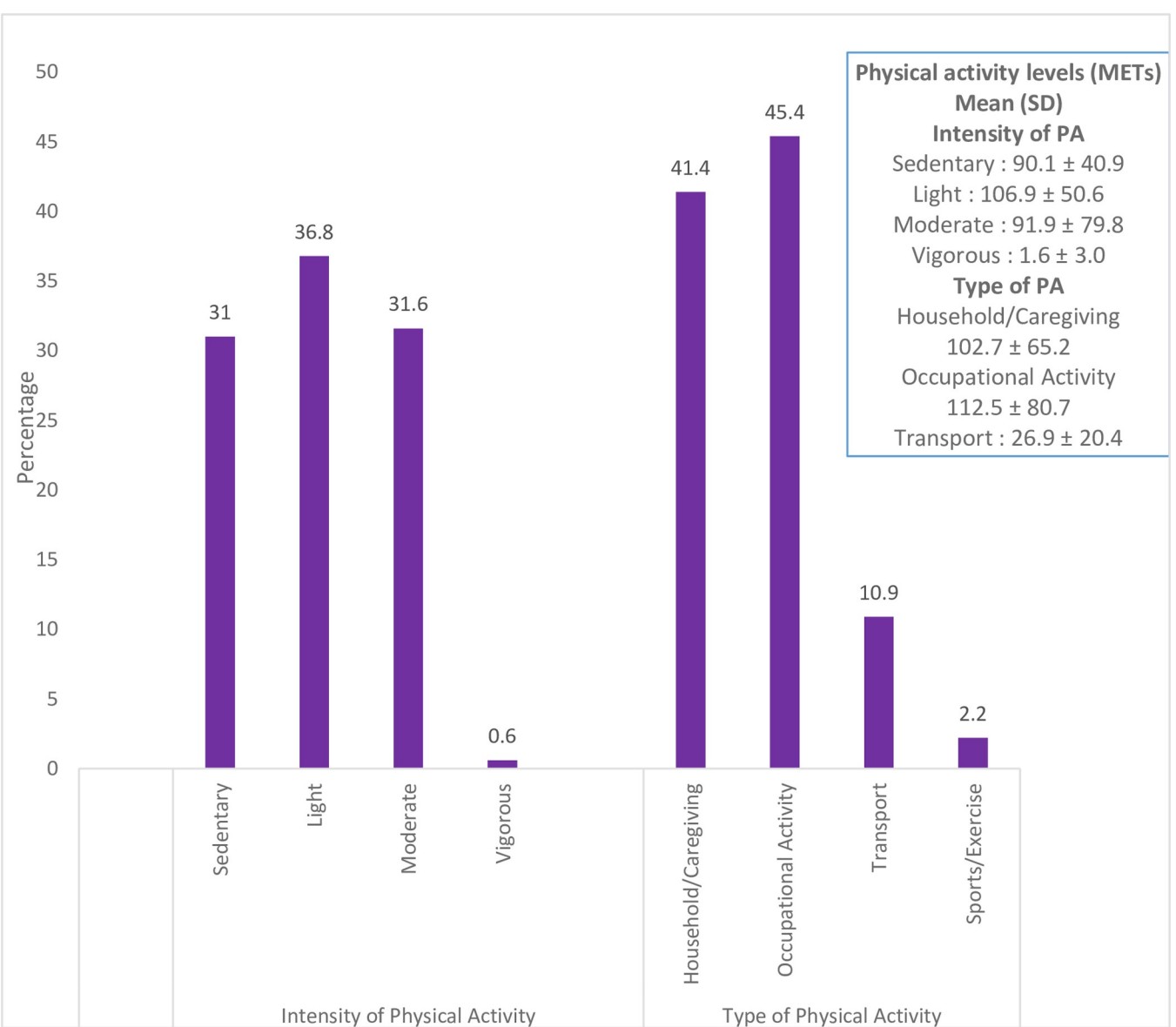

**Fig 1. Intensity and types of physical activity during pregnancy.**

## Pattern, types and duration of physical activity and sedentary behaviour among pregnant women (Figs 1–3)

The intensity and types of physical activity engaged in by the pregnant women are shown in Fig 1. The mean total energy expenditure was 290.5 ± 124.8 MET-h/week. Almost seventy per cent (67.8%) of total activity was low in intensity: sedentary (90.1 ± 40.9 MET-h/week) and light intensity activity 106.9 ± 50.6 MET-h/week (36.8%), followed by moderate intensity 91.9 ± 79.8 MET-h/week, and vigorous activity was rare (0.6%). The pattern of energy expenditure was: occupation-related activity (45.4%), household/caregiving activities (41.4%), and transport-related (10.9%), while sports or exercise were infrequent (2.2%). Fig 2 displays a box-plot that shows the duration of moderate-intensity activity per week in minutes by pregnant women. The mean duration was 26.3 ± 22.9 minutes. None of the individuals in the study met

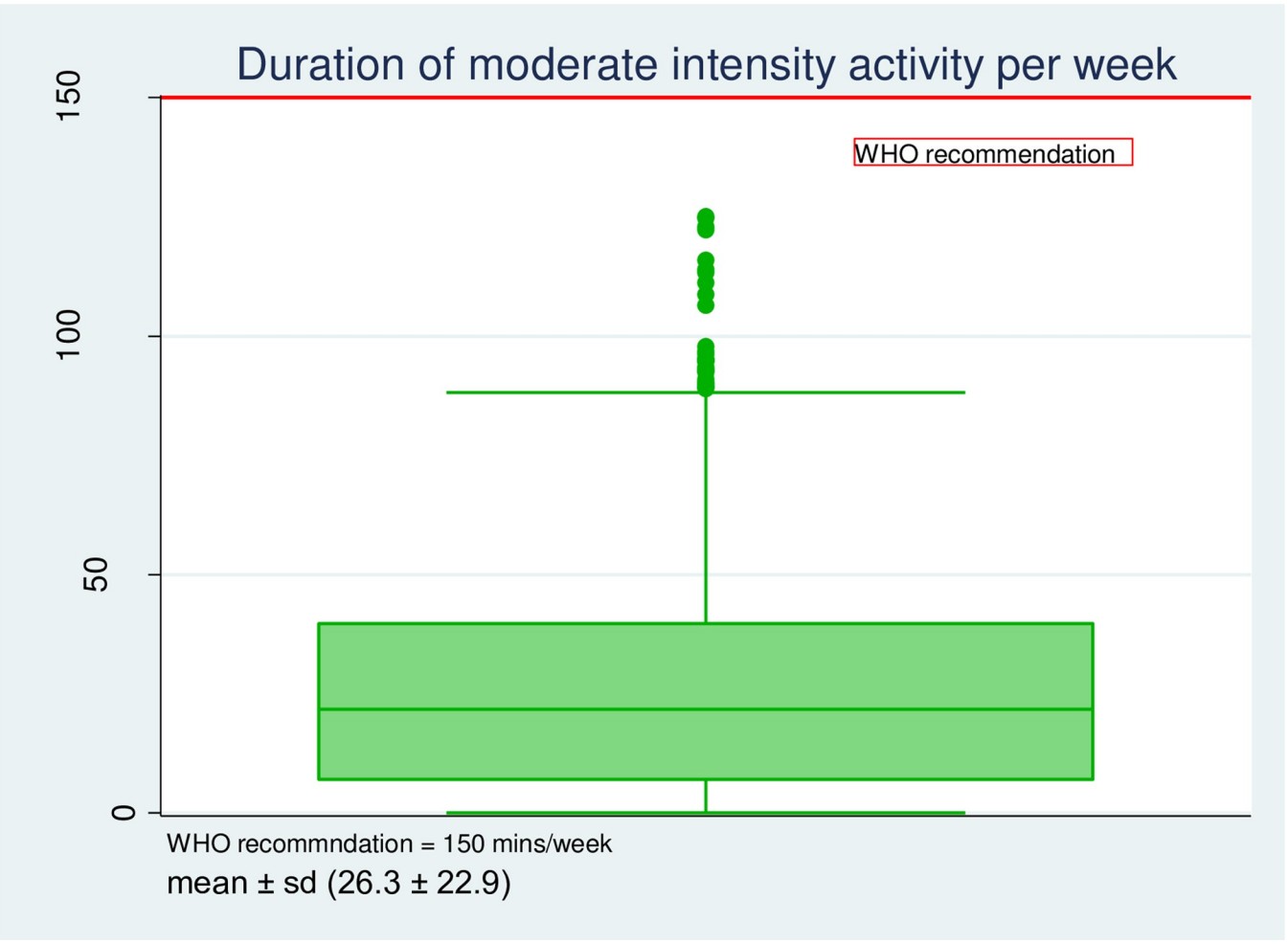

**Fig 2. Duration of moderate intensity activity among the pregnant women in Ibadan, Nigeria.**

the WHO recommendation for physical activity. Fig 3 shows the proportion of pregnant women that exceeded the recommended ST. About a third (29.1%) of the women exceeded the recommended limit pattern and duration (in hours) of sedentary behaviours. The mean daily sedentary time was 6.5 ± 4.2 hours.

## Discussion

Physical activity and sedentary behaviour have emerged as modifiable risk factors for adverse health outcomes in the general population and pregnant women. However, they have received very scant attention especially among pregnant women in Nigeria. Therefore, we assessed the pattern and factors associated with physical activity and sedentary behaviour of pregnant women using the PPAQ. The most crucial finding was the low level of physical activity among the study participants, as none of the pregnant women in this study met the WHO recommendation that pregnant women should engage in at least 150 minutes of moderate-intensity physical activity per week. Adeniyi *et al.* (2014) reported that in the same study setting, none of their study participants met the physical activity recommendation [34]. Other studies from different parts of the world have also documented low physical activity among pregnant women [35–37]. Very few pregnant women ever meet the recommended allowance for physical

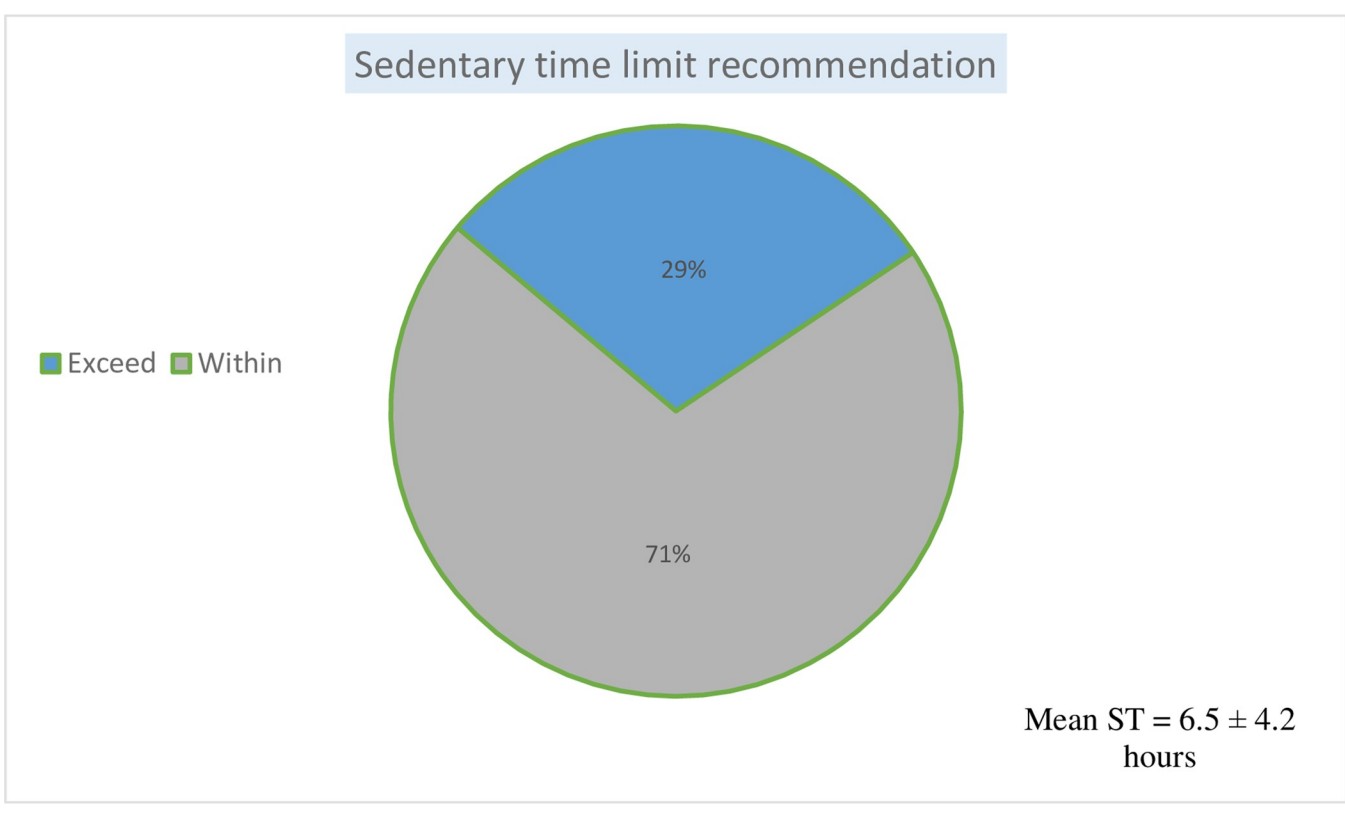

**Fig 3. Proportion of women that exceed the recommended sedentary time limit by pregnant women in Ibadan, Nigeria.**

activity. This study's average duration of moderate-intensity physical activity was 26.3 ± 22.9 minutes per week, implying that Nigerian women do not obtain sufficient benefits from physical activity during pregnancy. These benefits include improved cardiovascular fitness, reduced cardio-metabolic outcomes such as gestational diabetes mellitus, excessive gestational weight gain, hypertension, and enhanced mood [4–6]. The reason for the deficient level of physical activity may include the following: a lack of awareness of the benefits of physical activity by pregnant women and the health care providers, failure to incorporate and promote physical activity in routine maternal care services, the fear of complication especially miscarriage, lack of time, cultural inhibitions that encourage rest rather than physical activity during pregnancy [8,9]. It is necessary to explore the reasons for low physical activity levels in our environment in future studies using quantitative and qualitative methods. Zhang et al. (2014) reported that the fear of miscarriage was the most crucial reason for physical inactivity among urban Chinese pregnant women [35]. It is recommended that physical activity be encouraged among pregnant women during antenatal care.

Low-intensity physical activity was the study's most common form of physical activity, a finding corroborated by several researchers [34–38] as physical activity usually declines during pregnancy [36,39]. Hence, a light-intensity activity like walking (< 3.0 METs) is better than no activity. Walking has several maternal and fetal benefits and is the preferred mode of physical activity among pregnant women [40]. Besides, its intensity can be increased to moderate (3.0–6.0 METs) and even vigorous (> 6.0 METs) by increasing the pace. The maternal-fetal benefits of walking during pregnancy have been documented to include reduced risk of gestational diabetes mellitus, pre-eclampsia, unhealthy weight gain, and weight-related neonatal outcomes such as macrosomia, shoulder dystocia, and congenital anomaly [40]. Our maternal health

care services should incorporate exercise routines that encourage walking. On the other hand, the prominence of light intensity activity in our study may be due to the characteristics of our study population, which include urban residence and high-level education. Several African countries' ongoing epidemiological and nutritional transitions, especially cities, are associated with physical inactivity and sedentary occupations [29]. Hence this finding is not generalizable to the rural population. For example, a study that assessed physical activity levels during pregnancy in Northern Ethiopia [30] found that most women were involved in moderate-intensity physical activity almost every day, and 77.1% met the recommendation for physical activity. Notably, these women were of low socioeconomic status, performing household chores, and spent the highest energy on household activities (69.4 MET-h/weeks).

The correlates of inadequate physical activity were parity, occupation and religion. Other researchers have reported that parity, previous maternal history of miscarriage, mothers' education, maternal age, and employment were significantly associated with physical activity [35–37]. Women with higher parity had lower odds for inadequate physical activity in a dose-response fashion, i.e. "para 1–3" (AOR = 0.67) and para $\geq$ 4 (AOR = 0.57) compared with nulliparous women after controlling for confounders. A low level of physical activity among nulliparous women may result from inexperience, lack of information or misinformation, the notion that pregnancy is a time of rest rather than physical exercise, lower levels of household and caring duties compared to women of higher parities. Conversely, women with higher parity are likely to be engaged in more household and care-related activities. Apart from walking, household chores also allow pregnant women to take part in physical activity. Researchers have reported that household activities were the most accustomed activity among pregnant women [30,34]. Furthermore, women employed had a much lower likelihood (AOR = 0.23) of have inadequate physical activity compared with unemployed women. In this study, occupational-related physical activity was the most conversant physical activity reported by our study population. Probably due to the high-level education and the women mostly being in the working-class. The workplace also provides the opportunity for physical activity and energy expenditure during pregnancy. However, several occupations are now sedentary, requiring prolonged hours of sitting at work, associated with minimal energy expenditure. In this study, the average time spent sitting at work was 4.1 ± 2.7 hours, and about 60% of the women reported that their work required prolonged sitting.

Studies evaluating sedentary behaviour among the Nigerian pregnant population are lacking. In this study, sedentary behaviour was assessed using the PPAQ; sedentary time was also estimated to be spent sitting at work, using the computer, and watching television during the week and weekends. Earlier studies, mostly from high-income countries, had used various indicators to assess sedentary behaviour and time, for example, the time spent watching television [41,42], time sitting at work and other sedentary behaviours [43,44] or used objective measures [45,46]. Recently, the Sedentary Behaviour Research Network–a global collaboration of researchers and professionals on sedentary behaviour, developed consensus definitions and conceptual frameworks to be used in sedentary behaviour-related studies [15]. Additionally, the Canadian movement guideline for adults recommended that sedentary time be limited to eight hours or less with 24 hours in addition to adequate physical activity and sleep [33]. In this study, the estimated sedentary time was (6.5± 4.2) hours, and over a quarter of our study participants exceeded the eight hours limit for sedentary behaviour. Sedentary behaviour is a health challenge because it increases cardio-metabolic risk by reducing metabolic rate, contractility of muscles, glucose uptake, blood flow and so on [43,47]. It also increases the risk of adverse perinatal outcomes as gestational diabetes [48], excessive gestational weight gain [49], hypertension and deep vein thrombosis [21]. Hence, the need for targeted public health messages and antenatal education about the benefits of physical activity and the need to reduce

sedentary behaviour among pregnant women. However, some have documented not smoking [46], non-compliance with physical activity recommendations [50], maternal age and level of education [21] as factors associated with sedentary behaviour. For example, a US study which reported pregnant women spent 50% of their daily activity in sedentary behaviour also found that the odds of passive behaviour were lower among pregnant women who met the physical activity recommendation in their study population [46].

The factors associated with sedentary behaviour in the univariate analysis included parity, education, religion, and income. However, the multivariate analysis showed that high parity was protective of sedentary behaviour after controlling for confounding variables. The association between increasing parity and lower odds of sedentary behaviour is likely linked to expanded household and child caring activities. Tertiary level education and rising income, which are measures of socioeconomic status, had a direct association with sedentary behaviour. Tertiary education often affords high-paying, sedentary jobs with minimal energy expenditure. In this study, sedentary occupation in which participants majorly carried their duties in the sitting position was reported by 60% of our pregnant population. Jones et al. (2021), in a recent study, noted that primarily sitting occupation was associated with a higher odds of sedentary behaviour among pregnant women in their study [51]. Additionally, high income affords the means for sedentary entertainment and automated devices [52]. Few studies have explored the factors associated with sedentary behaviour [53], among pregnant women. However, some have documented not smoking [46], non-compliance with physical activity recommendations [50], maternal age and level of education [21] as factors associated with sedentary behaviour. For example, a US study which reported pregnant women spent 50% of their daily activity in sedentary behaviour also found that the odds of sedentary behaviour were lower among pregnant women who met the physical activity recommendation in their study population [46].

Our study fills a crucial but neglected gap in maternal health in Nigeria by examining pregnant women's patterns and correlates of physical activity and sedentary behaviours in Ibadan, Nigeria, using a large sample from the Ibadan Pregnancy Cohort Study in Nigeria. However, it has some limitations, which include a limited external validity to rural-dwelling women. Using a self-reported questionnaire may be associated with misclassification bias from under or over-reporting. Also, the study was conducted during early gestation, so the findings may not apply to late gestation. Further studies should explore the relationship between physical activity and pregnancy outcomes among Nigerian women.

## Conclusion and implication of the study

Pregnant women's physical activity and sedentary behaviours are emerging public health issues, especially in Nigeria. This study examined patterns and correlates of physical activity and sedentary behaviours of pregnant women in Ibadan, Nigeria, using the Pregnancy Physical Activity Questionnaire (PPAQ). The physical activity level was low as none of the pregnant women met the physical activity recommendation, and sedentary behaviour was prevalent. Parity and being gainfully employed were factors associated with physical activity, while parity, level of education and income were associated with sedentary behaviour. There is a need to emphasize the benefits of physical activity and limit sedentary behaviour among pregnant women in Nigeria.

## Acknowledgments

Special thanks to my research team for their dedication, support and hard work–research nurses, laboratory scientists, numerous research assistants Data personnel. We also wish to

appreciate the health workers–doctors, nurses, and clinic staff as well as the record staff of the various health facilities for their cooperation and support in the four facilities: University College Hospital, Adeoyo Maternity Teaching Hospital, Jericho Specialist Hospital, and Saint Mary Catholic Hospital Oluyoro, Ibadan. We appreciate the input of CARTA (Consortium for Advanced Research Training for Africa) for all its training, care, support, oversight, and funding and sponsorship efforts. Dr. Fagbamigbe's input into the manuscript is also appreciated.

## Author Contributions

**Conceptualization:** Ikeola A. Adeoye.

**Data curation:** Ikeola A. Adeoye.

**Formal analysis:** Ikeola A. Adeoye.

**Funding acquisition:** Ikeola A. Adeoye.

**Investigation:** Ikeola A. Adeoye.

**Methodology:** Ikeola A. Adeoye.

**Project administration:** Ikeola A. Adeoye.

**Resources:** Ikeola A. Adeoye.

**Software:** Ikeola A. Adeoye.

**Supervision:** Ikeola A. Adeoye.

**Validation:** Ikeola A. Adeoye.

**Visualization:** Ikeola A. Adeoye.

**Writing – original draft:** Ikeola A. Adeoye.

**Writing – review & editing:** Ikeola A. Adeoye.

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
