## [Decision Letter · Decision Letter 0]

13 Jul 2022

PGPH-D-22-00666

Pattern and correlates of physical activity and sedentary behaviors of pregnant women in Ibadan, Nigeria: Findings from the Ibadan pregnancy cohort study

Dear Dr. Adeoye,

Thank you for submitting your manuscript to PLOS Global Public Health. Your paper was reviewed by two external reviewers, who raised important points After careful consideration, we feel that it has merit but does not fully meet PLOS Global Public Health’s publication criteria as it currently stands. Therefore, we invite you to submit a revised version of the manuscript that addresses the points raised during the review process.

We look forward to receiving your revised manuscript.

Kind regards,

Julia Robinson

Executive Editor

Journal Requirements:

2. In the online submission form, you indicated that "The Ibadan Pregnancy Cohort Study is an ongoing study it is premature to put the data in the public domain now. Besides the datasets generated contain potentially identifying and confidential information. However, data sharing could be considered at a later time on reasonable request from the corresponding author if they meet the criteria for accessing confidential data.". 

Additional Editor Comments (if provided):

Reviewers' comments:

Reviewer's Responses to Questions

**Comments to the Author**

1. Does this manuscript meet PLOS Global Public Health’s publication criteria? Is the manuscript technically sound, and do the data support the conclusions? The manuscript must describe methodologically and ethically rigorous research with conclusions that are appropriately drawn based on the data presented.

Reviewer #1: Yes

Reviewer #2: Yes

2. Has the statistical analysis been performed appropriately and rigorously?

Reviewer #1: Yes

Reviewer #2: Yes

3. Have the authors made all data underlying the findings in their manuscript fully available (please refer to the Data Availability Statement at the start of the manuscript PDF file)?

Reviewer #1: No

Reviewer #2: Yes

4. Is the manuscript presented in an intelligible fashion and written in standard English?

Reviewer #1: Yes

Reviewer #2: Yes

5. Review Comments to the Author

Reviewer #1: A few english editorial issues, I think they will address them.

One section of the methods is too much, it should b given some sub titles to ensure the readers understand.

I have included that in my comments in the paper

Reviewer #2: Physical activity and sedentary behaviors of pregnant women is an emerging public health issue especially in Sub-Saharan Africa. Therefore, investigations that would result into increased physical activity and reduced sedentary behaviour among this population is valuable. Thus the manuscript presents a pertinent issue in a well writen manner. However, the following needs to be adressed by the authors:

• Chi square mentioned in the methodology, but missing in the result section.

• It is not clear which variables were adjusted for in coming up with AOR.

• Given the physical activity was found to be inadequate, the authors need to assess the correlates of inadequate physical activity instead of high level activity.

• Figure 1-3 are missing; moreover, not mentioned anywhere in the manuscript.

6. PLOS authors have the option to publish the peer review history of their article (what does this mean?). If published, this will include your full peer review and any attached files.

**Do you want your identity to be public for this peer review?** For information about this choice, including consent withdrawal, please see our Privacy Policy.

Reviewer #1: **Yes: **Geoffrey Babughirana

Reviewer #2: **Yes: **COLLINS OTIENO ASWETO

---

## [Decision Letter · Decision Letter 1]

16 Sep 2022

Pattern and correlates of physical activity and sedentary behaviors of pregnant women in Ibadan, Nigeria: Findings from the Ibadan pregnancy cohort study

PGPH-D-22-00666R1

Dear Dr Adeoye,

We are pleased to inform you that your manuscript 'Pattern and correlates of physical activity and sedentary behaviors of pregnant women in Ibadan, Nigeria: Findings from the Ibadan pregnancy cohort study' has been provisionally accepted for publication in PLOS Global Public Health.

Best regards,

Julia Robinson

Executive Editor

Reviewer Comments (if any, and for reference):

Reviewer's Responses to Questions

**Comments to the Author**

1. If the authors have adequately addressed your comments raised in a previous round of review and you feel that this manuscript is now acceptable for publication, you may indicate that here to bypass the “Comments to the Author” section, enter your conflict of interest statement in the “Confidential to Editor” section, and submit your "Accept" recommendation.

Reviewer #1: All comments have been addressed

2. Does this manuscript meet PLOS Global Public Health’s publication criteria? Is the manuscript technically sound, and do the data support the conclusions? The manuscript must describe methodologically and ethically rigorous research with conclusions that are appropriately drawn based on the data presented.

Reviewer #1: Yes

3. Has the statistical analysis been performed appropriately and rigorously?

Reviewer #1: Yes

4. Have the authors made all data underlying the findings in their manuscript fully available (please refer to the Data Availability Statement at the start of the manuscript PDF file)?

Reviewer #1: Yes

5. Is the manuscript presented in an intelligible fashion and written in standard English?

Reviewer #1: Yes

6. Review Comments to the Author

Reviewer #1: I have read the document again, I have made very minor comments for your consideration

7. PLOS authors have the option to publish the peer review history of their article (what does this mean?). If published, this will include your full peer review and any attached files.

**Do you want your identity to be public for this peer review?** For information about this choice, including consent withdrawal, please see our Privacy Policy.

Reviewer #1: **Yes: **Geoffrey Babughirana
